# Exploiting a Zoo of Checkpoints for Unseen Tasks

**Jiaji Huang**
Baidu Research
Sunnyvale, CA, 94089
huangjiaji@baidu.com

**Qiang Qiu**
School of Electrical and Computer Engineering
Purdue University, West Lafayette, IN, 47907
qqiu@purdue.edu

**Kenneth Church**
Baidu Research
Sunnyvale, CA, 94089
kennethchurch@baidu.com

## Abstract

There are so many models in the literature that it is difficult for practitioners to decide which combinations are likely to be effective for a new task. This paper attempts to address this question by capturing relationships among checkpoints published on the web. We model the space of tasks as a Gaussian process. The covariance can be estimated from checkpoints and unlabeled probing data. With the Gaussian process, we can identify representative checkpoints by a maximum mutual information criterion. This objective is submodular. A greedy method identifies representatives that are likely to "cover" the task space. These representatives generalize to new tasks with superior performance. Empirical evidence is provided for applications from both computational linguistics as well as computer vision[1].

## 1 Introduction

There are many model checkpoints published. For example, upon acceptance of this paper, the Huggingface repository[2] includes over 18,000 checkpoints trained for dozens of tasks (The tasks are tagged very coarsely. There will be many more if we tag in a fine-grained way). On the other hand, novel tasks can always arise, exciting tremendous interest in meta learning or learning to learn [1, 2, 3, 4]. This paper does not propose a new learning-to-learn method. Rather, we are interested in the question, "*Can we use the checkpoint zoo to build something that can adapt to unseen tasks?*"

To address the question, we have to understand how tasks are relevant to each other. A paraphrase of task relevance is task transferability [5]. Intuitively, transferring to a new task is easier if we warm start from a very relevant one. Efforts in this space originates from computer vision applications. Earlier work like Taskonomy [5] explicitly uses performance metrics (e.g., test accuracy) of transfer learning. Since then, multiple works have developed task descriptors to avoid running training and evaluation as Taskonomy. A few prominent examples are Task2vec [6], Attribution Map (A-Map) [7], and LEEP [8]. Very recently, Task2vec has also been extended to computational linguistics [9]. Each of these methods has its own pros and cons. However, these studies mainly aim at interpreting the performance of transfer learning [9, 10]. They mainly attack the question, "For a target task, which source task is the most relevant?". It is different from the question we are interested in.

Understanding the relation among tasks is crucial for meta learning and transfer learning. It is known that dissimilar tasks can impose significant challenges for these learning frameworks [11, 12]. Aware

---

[1]All results can be reproduced using code at https://github.com/baidu-research/task_space
[2]https://huggingface.co/models

35th Conference on Neural Information Processing Systems (NeurIPS 2021).

of this fact, several recent works [13, 14] develop generalization error bounds for learning new tasks. In particular, they address the question, "how many tasks has to be seen, so that a meta-trained model can generalize to new tasks?". These works often assume that there is a shared function, mapping data to representations for all tasks. Then they propose their measure of task diversity, which is further employed to bound the generalization error for a new task. These bounds suggest that the seen tasks must be sufficiently diversified to "cover" all components that may be used for a future task. However, it is not obvious how to compute these diversity measures in practice.

Back to our problem, a zoo of checkpoints solve a collection of seen tasks. We are interested in combining/fusing a subset of them. Motivated by the theoretical works [13, 14], we want the selected checkpoints (essentially the tasks they solve) to be diversified. In a perhaps tangential problem [15], the authors consider where to place sensors in a room such that they are the most informative about the undeployed locations. They show that the placement should be diversified and well represent the landscape. To this end, we could draw an analogy. Consider the task space as a room, and checkpoints as sensors. The unseen tasks are locations where no sensor is placed but we wish to infer their information. In later sections, we shall see how this analogy helps us design a strategy for selecting checkpoints.

A relevant concept to our work is model ensemble, which has a long history [16], and still actively being studied recently [17]. However, model ensemble usually assumes a shared output space for all component models, therefore no notion of multiple (seen or unseen) tasks. Another relevant topic is feature selection [18]. Indeed, if we consider each checkpoint a feature extractor, picking checkpoints amount to identifying useful features. However, feature selection methods usually aim at finding the most relevant features for a target variable. In other words, they only target at a single seen task.

## 2  Motivating the Problem

Consider a zoo of $n$ checkpoints, where the $i$-th checkpoint is trained for the $i$-th task. In this paper, we define the $i$-th task as a joint distribution $p_i(\mathbf{x}, \mathbf{y})$ to be fit to, where $\mathbf{x} \in \mathcal{X}$ is input and $\mathbf{y} \in \mathcal{Y}_i$ is target output. The $i$-th checkpoint is trained on a dataset $\mathcal{D}_i$ drawn from $p_i$. $\mathcal{D}_i$ is often proprietary. Or due to non-trivial preprocessing, it is very difficult to obtain the data exactly as it was. The tasks are assumed to share the same input space, but differ in their target spaces. This is a natural assumption. Consider a zoo of convolutional networks that handle various computer vision tasks [5], or a collection of huggingface transformers for multiple linguistic tasks. The $\mathcal{X}$ is a space of images for the former case, and text for the later.

Suppose there is a universe of (possibly infinitely many) tasks, drawn from a "hyper-distribution" $\mathcal{P}$. That is $(p_1, \ldots, p_n, p_{n+1}, \ldots) \sim \mathcal{P}$. Modeling a family of tasks as "distribution over distributions" [1, 19] has a long history in learning-to-learn literature. The $n$ checkpoints in the zoo have distilled information from $p_1, \ldots, p_n$, and we are interested in building a "stronger" model from them that can easily generalize to unseen tasks $(p_{n+1}, \ldots)$. It should be emphasized that these new tasks are not revealed to us before we build the "stronger" model.

To be more specific, let the $i$-th checkpoint be in the (commonly seen) form, $\mathbf{h}_i \circ \mathbf{f}_i : \mathcal{X} \mapsto \mathcal{Y}_i$, where $\mathbf{f}_i$ is a feature extractor. $\mathbf{h}_i$ is a task-specific "head" for the $i$-th task, and $\circ$ denotes function composition. $\mathbf{h}_i$ is often a "simple" function, e.g., softmax classifier. Note that $\mathbf{h}_i$ *may not necessarily be available*. This is true for many checkpoints in huggingface repository. We are interested in selecting a subset $\mathcal{S}$ of checkpoints from the zoo *before seeing any new task* $p_{n+1}, \ldots$, and building a "stronger" feature extractor by concatenating the selected $\mathbf{f}_i$'s, $\mathbf{f} \triangleq \otimes_{i \in \mathcal{S}} \mathbf{f}_i$. The $\mathbf{f}$ should be quickly adaptable to new tasks $p_{n+1}, \ldots$. In other words, given some small training data for a new task, we want a task-specific "head" on top of the $\mathbf{f}$ to be "easily" learned and work well on test data.

Of course, this approach should have some budget constraints. Indeed, when $n$ is big, concatenating all the $\mathbf{f}_i$'s may be costly in computation, and result in overfitting. At this point, one may propose to use feature selection methods like min-Redundancy-Max-Relevance (mRMR) [18]. That is, to pick the $\mathbf{f}_i$'s that are the most informative for a new task, meanwhile least redundant mutually. However, in our case, no new task is revealed to us before selecting. Thus we cannot simply apply mRMR.

Having ruled out feature selection methods, we start to re-think the problem by modeling the space of tasks. To this end, we may make an analogy to optimal sensor placement [15]. In that application, one is supposed to pick the best locations to place sensors, so that their readouts are

the most informative about the unsensed locations. In information theoretic language, this would be $\max I(\text{sensed location}; \text{unsensed location})$, where $I(\cdot, \cdot)$ is mutual information. In our problem, each checkpoint handles its specific task. It can be considered as a "sensor" placed at a task. Then, the checkpoint zoo is a "pilot deployment" that places many "sensors" on grids of the task space, which can reveal how the tasks are related. Selecting checkpoints is essentially picking a few seen tasks that may be the most informative about the unseen ones. In the following sections, we concretize the above idea by *modeling and exploiting the task space*.

## 3 Modeling the Task Space

Theoretical works [1, 19, 13] often do not specify a form for the hyper-distribution $\mathcal{P}$. In this work, we assume $\mathcal{P}$ a Gaussian process. So a next step is to define the covariance function, $\kappa(p_i, p_j)$, for $\mathcal{P}$. In other words, we want to come up with a similarity measure between the two distributions $p_i$ and $p_j$. Moreover, by definition, the $\kappa$ has to be Positive Semi-Definite (PSD). In the following, we first rule out several options that are less applicable. Then we propose our approach.

### 3.1 A Taxonomy of Task Similarities

If we had access to data samples $\mathcal{D}_i$, and $\mathcal{D}_j$, a natural idea would be comparing $p_i$ and $p_j$ directly. The difficulty is that the output spaces $\mathcal{Y}_i$ and $\mathcal{Y}_j$ are not directly comparable. For example, $\mathcal{Y}_i$ may be the input images' class labels, whereas $\mathcal{Y}_j$ may be the foreground to be segmented. Therefore, straightforward measures like KL divergence cannot be applied. To address this issue, Geometric Dataset Distance (GDD) [20] proposes to estimate Wasserstein distance between $p_i$ and $p_j$, by carefully designing the "transport cost" between $\mathcal{Y}_i$ and $\mathcal{Y}_j$ using $\mathcal{D}_i$ and $\mathcal{D}_j$. However, $\mathcal{D}_i$ and $\mathcal{D}_j$ are often proprietary. For example, the data to train huggingface transformers are often preprocessed with various tools. It is very hard (if not impossible) to recover the exact same $\mathcal{D}_i$.

Other approaches treat $\kappa(p_i, p_j)$ as the transferability from the $i$-th task to the $j$-th. Taskonomy [5] empirically computes that by running training and testing for this $i$-to-$j$ transfer learning problem. Doing that for all $(i, j)$ pairs is very costly. Besides, non-trivial normalizations have to be applied as performance metrics can differ for different tasks. To circumvent this brute-force process, people have come up with vector representations of tasks, reducing tasks similarities to distances in this vector space. For example, Task2vec [6] argues that Fisher information matrix (essentially gradient w.r.t. parameters in the checkpoint) is an indicative representation for the corresponding task. However, it assumes that all checkpoints have the same model configuration, which is a very strong assumption and need not be satisfied in practice. Another work, A-Map [7] is based on the intuition that similar tasks should have similar saliency map, which is the gradient of (pooled) feature w.r.t. input. Therefore, it only requires some unlabelled probing data, and model architectures can be different. However, the back-propagation can be more costly. Further more, to compare two saliency maps, their inputs have to be the same. This constraint can be violated for NLP applications, as the same input text is often mapped to two different sequences of embedding vectors using two checkpoints.

Recently, NCE [21] and LEEP [8] measure task transferability by estimating an approximation of conditional entropy, $H(\mathbf{y}_j|\mathbf{y}_i) = H(\mathbf{y}_j) - I(\mathbf{y}_i; \mathbf{y}_j)$. Ignoring $H(\mathbf{y}_j)$ and focusing on the mutual information term, we can understand these two methods as measuring the dependency between two target spaces $\mathcal{Y}_i$ and $\mathcal{Y}_j$. However, to estimate $H(\mathbf{y}_j|\mathbf{y}_i)$, data labels must be available.

For clarity, we list the discussed methods in Tab. 1. Also, the last row of Tab. 1 highlights some properties a desired method should have. We can see various reasons why these existing methods are not applicable. Nevertheless, LEEP [8] provides us with a principled way of understanding the relationship between two tasks, *i.e.*, measuring dependency between their target spaces. We next derive another approximation to this dependency, which is a desired method.

### 3.2 Proposed Approach

Following the setup in NCE [21] and LEEP [8], we also assume $m$ input samples $\mathbf{X} \triangleq [\mathbf{x}^1, \ldots, \mathbf{x}^m]$, with target outputs $\mathbf{Y}_i \triangleq [\mathbf{y}_i^1, \ldots, \mathbf{y}_i^m]$ in the $i$-th task, and $\mathbf{Y}_j \triangleq [\mathbf{y}_j^1, \ldots, \mathbf{y}_j^m]$ in the $j$-th task. We wish to measure the dependency between $\mathbf{Y}_i$ and $\mathbf{Y}_j$. One approach is Kernel Alignment

Table 1: Candidate methods to measure task relevance. Those requiring access to $\mathcal{D}_i$ or $\mathcal{D}_j$ are not applicable in our setting. Those requiring back-propagation are costly. $\kappa \succeq 0$ also rules out some candidates. In contrast, the last row highlights a desired method.

| Method | task $i$ | | task $j$ | | back-prop | Probing data (unlabelled) | $\kappa \succeq 0$ |
|---|---|---|---|---|---|---|---|
| | checkpoint $i$ | data $\mathcal{D}_i$ | checkpoint $j$ | data $\mathcal{D}_j$ | | | |
| GDD [20] | ✓ | ✓ | ✓ | ✓ | ✗ | ✗ | ✓ |
| Taskonomy [5] | ✓ | ✓ | ✓ | ✓ | ✓ | ✗ | ✗ |
| Task2vec [6] | ✓ | ✓ | ✓ | ✓ | ✓ | ✗ | ✗ |
| A-Map [7] | ✓ | ✗ | ✓ | ✗ | ✓ | ✓ | ✓ |
| LEEP [8] | ✗ | ✓ | ✓ | ✗ | ✗ | ✗ | ✗ |
| desired method | ✓ | ✗ | ✓ | ✗ | ✗ | ? | ✓ |

(KA) [22, 23]. In the case of linear kernels, KA can be calculated as

$$\mathrm{KA}(\mathbf{Y}_i, \mathbf{Y}_j) = \frac{\langle \mathbf{Y}_i^\top \mathbf{Y}_i, \mathbf{Y}_j^\top \mathbf{Y}_j \rangle}{\langle \mathbf{Y}_i^\top \mathbf{Y}_i, \mathbf{Y}_i^\top \mathbf{Y}_i \rangle \cdot \langle \mathbf{Y}_j^\top \mathbf{Y}_j, \mathbf{Y}_j^\top \mathbf{Y}_j \rangle}.$$

To give some intuitions of the above, let columns of $\mathbf{Y}_i$ and $\mathbf{Y}_j$ be one-hot. Then $\mathbf{Y}_i^\top \mathbf{Y}_i$ (same for $\mathbf{Y}_j^\top \mathbf{Y}_j$) is a binary matrix who assigns 1 for two inputs with the same label, and 0 otherwise. KA measures the dependency between the two tasks by cosine similarity of their assignment patterns.

However, we do not have access to the data labels. Moreover, sometimes the task-specific heads are unavailable, so even a prediction of the labels (denoted as $\hat{\mathbf{Y}}_i$ and $\hat{\mathbf{Y}}_j$) cannot be obtained. We tackle this difficulty by resorting to unsupervised "probing" data $[\mathbf{x}^1, \ldots, \mathbf{x}^m]$. Note that this $[\mathbf{x}^1, \ldots, \mathbf{x}^m]$ need not be the inputs that train the checkpoint(s). Regardless of the availability of $\mathbf{h}_i(\cdot)$ and $\mathbf{h}_j(\cdot)$, we can extract feature representations by

$$\mathbf{F}_i \triangleq [\mathbf{f}_i(\mathbf{x}^1), \ldots, \mathbf{f}_i(\mathbf{x}^m)], \text{ and } \mathbf{F}_j \triangleq [\mathbf{f}_j(\mathbf{x}^1), \ldots, \mathbf{f}_j(\mathbf{x}^m)].$$

For notation easiness, in the following, we may write $\mathbf{f}_i^a = \mathbf{f}_i(\mathbf{x}^a)$ and $\mathbf{f}_j^a = \mathbf{f}_j(\mathbf{x}^a)$ for any input $\mathbf{x}^a$.

Then we use

$$\mathrm{KA}(\mathbf{F}_i, \mathbf{F}_j) = \frac{\langle \mathbf{F}_i^\top \mathbf{F}_i, \mathbf{F}_j^\top \mathbf{F}_j \rangle}{\langle \mathbf{F}_i^\top \mathbf{F}_i, \mathbf{F}_i^\top \mathbf{F}_i \rangle \cdot \langle \mathbf{F}_j^\top \mathbf{F}_j, \mathbf{F}_j^\top \mathbf{F}_j \rangle}, \tag{1}$$

as an approximation to $\mathrm{KA}(\mathbf{Y}_i, \mathbf{Y}_j)$.

A natural question to ask is how good this approximation is. We assure ourselves by the following reasoning. Let

$$\hat{\mathbf{Y}}_i = [\hat{\mathbf{y}}_i^1, \ldots, \hat{\mathbf{y}}_i^m] \triangleq [\mathbf{h}_i(\mathbf{f}_i^1), \ldots, \mathbf{h}_i(\mathbf{f}_i^m)], \text{ and } \hat{\mathbf{Y}}_j = [\hat{\mathbf{y}}_j^1, \ldots, \hat{\mathbf{y}}_j^m] \triangleq [\mathbf{h}_j(\mathbf{f}_j^1), \ldots, \mathbf{h}_j(\mathbf{f}_j^m)],$$

be the predictions made for each task (Although $\hat{\mathbf{Y}}_i$ may not be obtained as $\mathbf{h}_i$ is not available. Same for $\hat{\mathbf{Y}}_j$). If each checkpoint works sufficiently well for its own task, then $\hat{\mathbf{Y}}_i$ is very "close" to $\mathbf{Y}_i$, so is $\hat{\mathbf{Y}}_j$ to $\mathbf{Y}_j$. Thus $\mathrm{KA}(\hat{\mathbf{Y}}_i, \hat{\mathbf{Y}}_j)$ well approximates $\mathrm{KA}(\mathbf{Y}_i, \mathbf{Y}_j)$.

It remains to validate if $\mathrm{KA}(\mathbf{F}_i, \mathbf{F}_j)$ well approximates $\mathrm{KA}(\hat{\mathbf{Y}}_i, \hat{\mathbf{Y}}_j)$. The only gap between $\mathbf{F}_i$ and $\hat{\mathbf{Y}}_i$ is a task-specific head $\mathbf{h}_i(\cdot)$, likewise, $\mathbf{h}_j(\cdot)$ for $\mathbf{F}_j$ and $\hat{\mathbf{Y}}_j$. We argue that many task-specific architectures are "well conditioned" [24, 25], which guarantees that pairwise inner products are not distorted too much. This property further results in $\mathrm{KA}(\mathbf{F}_i, \mathbf{F}_j) \approx \mathrm{KA}(\hat{\mathbf{Y}}_i, \hat{\mathbf{Y}}_j)$. To be more concrete, consider a simple special case, where $\mathbf{h}_i(\mathbf{f}) = \mathbf{M}_i \mathbf{f}$ and $\mathbf{h}_j(\mathbf{f}) = \mathbf{M}_j \mathbf{f}$ are linear mappings. By the "well-conditioned" assumption, we have $\mathbf{M}_i^\top \mathbf{M}_i = c\mathbf{I}$ and $\mathbf{M}_j^\top \mathbf{M}_j = d\mathbf{I}$ for some $c, d > 0$. Then it is easy to show that $\mathrm{KA}(\mathbf{F}_i, \mathbf{F}_j) = \mathrm{KA}(\hat{\mathbf{Y}}_i, \hat{\mathbf{Y}}_j)$.

Last but not least, we have the following guarantee on positive definiteness.

**Proposition 1.** *Let $\kappa(i, j) = KA(\mathbf{F}_i, \mathbf{F}_j)$, then $\kappa \succeq 0$.*

We are not the first to use kernel alignment to understand deep nets. [26] proposes to use linear and RBF Kernel alignment to measure similarity between representations. Although the methodologies appear similar, the goals are quite different. [26] attempts to understand representation similarity, an open-ended problem. In contrast, we are explicitly interested in measuring the dependency between two tasks' output spaces. And it turns out that KA between their respective features is a sensible surrogate. Moreover, Proposition 1 enables us to construct a Gaussian process on the task space.

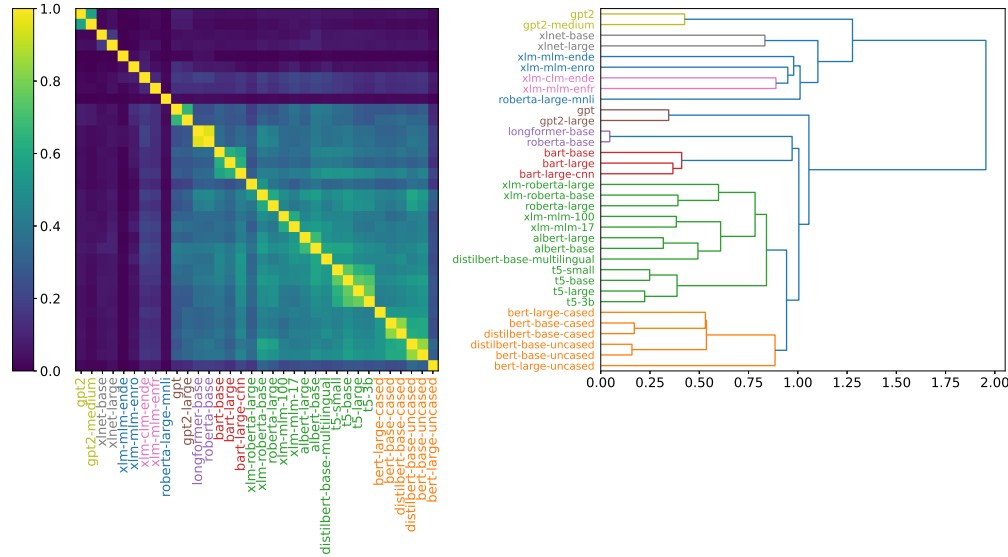

Figure 1: Left: $\kappa$ matrix computed using 34 checkpoints from huggingface. Note that by definition, the diagonal is all-one. Right: hierarchical clustering on $\kappa$. Colors denote clusters. Key observations: 1) Similar names usually share colors, e.g., bert-∗ checkpoints are orange and t5-∗'s are green, as they respectively handle the same task. 2) A checkpoint fine-tuned on another task can become very different, e.g., roberta-large-mnli against roberta-large.

### 3.3 Example: Characterize Linguistic Task Space using Huggingface Checkpoints

We showcase a concrete example using 34 checkpoints from huggingface. They are trained for various tasks (or their combinations). For example, the bert-∗ [27] checkpoints are trained for masked language modeling and next sentence prediction. t5-∗'s are for text-to-text generation. Complete details of these checkpoints and their respective tasks can be found in supplementary material. Note also that multiple checkpoints may target at the same task. These checkpoints are usually named with the same prefix. For example, the t5-∗'s only differ by their architecture configurations (e.g., number of layers). In analogous to sensor placement, this amounts to a "pilot deployment" that places multiple checkpoints at the same location in the task space. It is somewhat redundant to do so. However, they grant us a chance to do sanity check on the estimation of $\kappa$. That is, the checkpoints corresponding to the same task should have large $\kappa(i,j)$ values.

We input training set of wikitext2 as probing data, and extract the contextualized word embeddings after penultimate layer. So task specific layers like softmax classifiers are ignored. Note that these checkpoints employ different tokenizers, often resulting in the same word being split into different multiple sub-words. We adopt the strategy in [28], taking the representation of a word as that of its last sub-word's. Now columns in $\mathbf{F}_i$ are contextualized word representations from the $i$-th checkpoint. Then we can compute $\mathrm{KA}(\mathbf{F}_i, \mathbf{F}_j)$ for each pair of checkpoints to get the $\kappa$ matrix. To visualize the relationship between these checkpoints (thus the tasks they target), we apply hierarchical clustering[3] to the $\kappa$ matrix and visualize a dendrogram in fig. 1.

In fig. 1, we observe that checkpoints with the same prefix are usually close, e.g., the t5-∗'s. This is less surprising, since obviously they are trained for the same task. A more surprising discovery is that longformer-base [29] is very similar to roberta-base [28]. A closer check of their documentation informs us that they are both trained for masked language modeling, although longformer-base innovated a novel attention mechanism. On the other hand, adapting a checkpoint to another task can result in significant difference, e.g., roberta-large-mnli against roberta-large. Therefore, the $\kappa$ indeed reflects our intuitions on tasks.

Finally, there are also a few counter-intuitive cases. For example, gpt is very different from gpt2, but they are both trained for causal language modeling. We conjecture it is their different tokenizers that drive $\mathrm{KA}(\mathbf{F}_i, \mathbf{F}_j)$ to be small. A full understanding of this observation is deferred to future study.

---

[3]We use scipy functions: linkage with method="ward", and dendrogram with color_threshold=0.9.

### 3.4 Robustness of the Estimated Covariance

Estimating $\kappa$ relies on probing data. Therefore, it is natural to ask, how robust $\kappa$ is against size of the probing data. In addition, we have implicitly simplified and assumed that the probing data, as well as the inaccessible training inputs for all tasks, have the same distribution on $\mathcal{X}$.

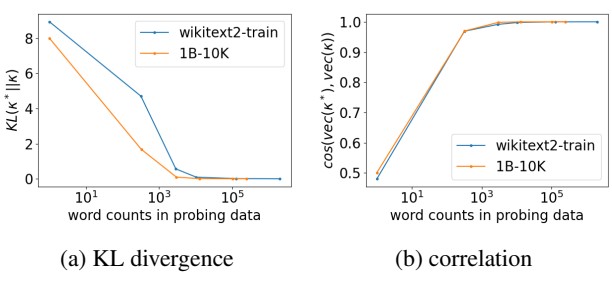

(a) KL divergence    (b) correlation

Figure 2: Measure the difference between $\kappa$ (estimated from a subset of probing data) and $\kappa^*$ (estimated from full probing data). $\kappa$ converges quickly to $\kappa^*$.

However, in practice, the input distribution may have domain shift from task to task, and from task to probing data as well. Hence, we also want to check whether $\kappa$ is stable against different genre of probing text.

We denote the covariance estimated in the previous section as $\kappa^*$. Now we estimate $\kappa$ using various subsets of the wikitext2 training set. It is guaranteed that any smaller subset is included in a bigger subset. We want to see how quickly these $\kappa$'s approach $\kappa^*$, as size of probing data increases. The same experiment is also repeated on another corpora, 10K sentences taken from billion-word benchmark, dubbed 1B-10K. This benchmark is created from news crawl, a different genre from wiki.

To compare $\kappa$ against $\kappa^*$, we measure the KL-divergence between two Gaussian distributions with $\kappa^*$ and $\kappa$ as their respective covariances. Another simple measure is the cosine similarity between vectorized $\text{vec}(\kappa)$ and $\text{vec}(\kappa^*)$. We plot $KL(\kappa^*||\kappa)$ and $\cos(\text{vec}(\kappa^*), \text{vec}(\kappa))$ against the number of words in the probing text, shown in fig. 2a and 2b respectively. For both plots, the left most part of the x-axis stands for zero probing data. In this case, it is natural to assume $\kappa$ an identity matrix, *i.e.*, all checkpoints (thus the tasks they were trained for) are independent.

In fig. 2, we observe that as the size of probing data increases, KL divergences quickly converge to 0, while the cosines quickly converge to 1. In addition, let $\kappa^*_{\text{wiki}}$ be the covariance estimated from full wikitext2 training set, and $\kappa^*_{\text{1B}}$ the covariance from full 1B-10K. It turns out that $KL(\kappa^*_{\text{wiki}}||\kappa^*_{\text{1B}}) = 0.76$, not a large value considering the range of KL-divergences in fig. 2a. On the other hand, their cosine similarity is as high as 0.98. To conclude, the estimate of covariance function is reasonably robust against size and genre of probing data.

## 4 Exploiting the Task Space

We have characterized the task space using a zoo of checkpoints. Now we are ready to selecting checkpoints. The idea is to pick seen tasks that are the most informative about the the task space. Again, remember that this is conducted before any new task is revealed to us.

Suppose we have a budget of using $K$ checkpoints. Let $\mathcal{Z}$ (with cardinally $n$) be the zoo of all checkpoints, and $\mathcal{S}$ the set of selected checkpoints. We want to maximize the mutual information between the tasks solved by $\mathcal{S}$, and the remaining of the task space. With some abuse of notation, we denote this mutual information as $I(\mathcal{S}; \mathcal{Z} - \mathcal{S})$. Here $\mathcal{S}$ should be understood as the tasks corresponding to the checkpoints in $\mathcal{S}$. We optimize the following constrained objective,

$$\max_{\mathcal{S} \subset \mathcal{Z}: |\mathcal{S}| = K} I(\mathcal{S}; \mathcal{Z} - \mathcal{S}). \tag{2}$$

The above a combinatorial optimization problem. Enumerating all cases is infeasible. Instead, a heuristic is to use greedy search. Starting from an empty $\mathcal{S}$, each time we include a checkpoint $i$ that brings the largest gain to the objective. In specific, denote $\bar{\mathcal{S}} = \mathcal{Z} - \mathcal{S}$, $\mathcal{S}_i = \mathcal{S} \cup \{i\}$ and $\bar{\mathcal{S}}_i = \mathcal{Z} - (\mathcal{S} \cup \{i\})$. The gain $\delta_i$ can be calculated as,

$$\begin{aligned}
\delta_i &= I(\mathcal{S}_i; \bar{\mathcal{S}}_i) - I(\mathcal{S}; \bar{\mathcal{S}}) \\
&= [H(\mathcal{S}_i) - H(\mathcal{S}_i|\bar{\mathcal{S}}_i)] - [H(\mathcal{S}) - H(\mathcal{S}|\bar{\mathcal{S}})] \\
&= [H(\mathcal{S}_i) - H(\mathcal{Z}) + H(\bar{\mathcal{S}}_i)] - [H(\mathcal{S}) - H(\mathcal{Z}) + H(\bar{\mathcal{S}})] \\
&= H(\{i\}|\mathcal{S}) - H(\{i\}|\bar{\mathcal{S}}_i)
\end{aligned} \tag{3}$$

The first term implies that $i$ should surprise the current $\mathcal{S}$ the most, thus incorporating more information. Meanwhile, the second term suggests that $i$ should be representative of the remaining of the task space, thus avoiding outliers.

Since we assume a Gaussian process on the task space, any subset of $\mathcal{Z}$ is Gaussian distributed. And the two terms in Eq. (3) can be easily calculated. Specifically, let $\kappa(\mathcal{S}, \mathcal{S})$ denote the covariance for subset $\mathcal{S}$, $\kappa(i, \mathcal{S})$ the row vector of cross-covariance between $\{i\}$ and $\mathcal{S}$, and $\kappa(\mathcal{S}, i)$ its transpose. Then,

$$\delta_i = H(\{i\}|\mathcal{S}) - H(\{i\}|\bar{S}_i) = \frac{1}{2} \ln \frac{1 - \kappa(i, \mathcal{S})\kappa(\mathcal{S}, \mathcal{S})^{-1}\kappa(\mathcal{S}, i)}{1 - \kappa(j, \bar{S}_i)\kappa(\bar{S}_i, \bar{S}_i)^{-1}\kappa(\bar{S}_i, i)} \tag{4}$$

The greedy selection process is summarized in Algorithm 1. Denote $\mathcal{S}(K)$ as the set of selected checkpoints for a $K$ value. Algorithm 1 guarantees that for any $K_1 < K_2$, $\mathcal{S}(K_1) \subsetneq \mathcal{S}(K_2)$.

---

**Algorithm 1** Maximum Mutual Information (MMI) based Selection of Checkpoints

---

**Input:** Checkpoint zoo $\mathcal{Z}$ with cardinality $n$, covariance matrix $\kappa \in \mathbb{R}^{n \times n}$,
   number of checkpoints to pick, $K$
**Output:** a set $\mathcal{S}$ of selected checkpoints, where $|\mathcal{S}| = K$
 1: $\mathcal{S} \leftarrow \varnothing$
 2: **for** $k = 1, \ldots, K$ **do**
 3:    **for** $i \in \mathcal{Z} - \mathcal{S}$ **do**
 4:       Compute information gain $\delta_i$ using Eq. (4)
 5:    **end for**
 6:    $i^* \leftarrow \arg\max \delta_i$
 7:    $S \leftarrow \mathcal{S} \cup \{i^*\}$
 8: **end for**

---

### 4.1 Discussion on Quality of the Greedy Approximation

It is natural to ask how good the greedy method is. At this point, we need to review the definition of submodular functions. A set function $f(\mathcal{S})$ is submodular if for any sets $\mathcal{S} \subseteq \mathcal{S}' \subseteq \mathcal{Z}$ and $i \in \mathcal{Z} - \mathcal{S}'$, $f(\mathcal{S} \cup \{i\}) - f(\mathcal{S}) \geq f(\mathcal{S}' \cup \{i\}) - f(\mathcal{S}')$. It is not hard to show that the following holds.

**Proposition 2.** $I(\mathcal{S}; \mathcal{Z} - \mathcal{S})$ is a submodular function of $\mathcal{S}$.

Another useful concept is monotonic function. A function $f(\mathcal{S})$ is monotonic if for any $i \in \mathcal{Z}$, $f(\mathcal{S} \cup \{i\}) \geq f(\mathcal{S})$. For monotonic submodular functions, it is known that a greedy method is guaranteed to achieve an objective no smaller than $1 - 1/e$ of the optimal one [30]. In our case, $I(\mathcal{S}; \mathcal{Z} - \mathcal{S})$ is submodular but not monotonic. As $I(\mathcal{Z}; \mathcal{Z} - \mathcal{S}) = 0$, indicating that when set $\mathcal{S}$ is sufficiently large, further enlarging $\mathcal{S}$ will reduce mutual information.

It thus becomes less clear if the $(1 - 1/e)$ guarantee still holds. In optimal senor placement [15], the authors show that if the covariance is estimated for sufficiently many fine-grained grids in the room, then the mutual information is monotonic up to a reasonably big $K$. This may suggest that our $\kappa$ has to be estimated for sufficiently many tasks. In other words, the checkpoint zoo must be rich enough. Nevertheless, it is hard to come up with a rigorous guarantee like [15], as assumptions made in 2D physical space does not necessarily transfer to an abstract task space. However, in experiments, we have found that Algorithm 1 always works in a monotonic regime.

## 5 Experiments

In the previous sections, we have presented two key components, estimation of $\kappa$ and MMI based selection of checkpoints. In this section, we experiment with the two components combined. First, we apply algorithm 1 to the $\kappa$ estimated in section 3.3, and show its effectiveness on multiple linguistic tasks. The baseline we compare against is random selection of checkpoints, and single commonly adopted checkpoint, e.g., bert-base-uncased. Then we extend to image classification tasks. Again we observe constant improvements over random picks, and other straightforward alternative.

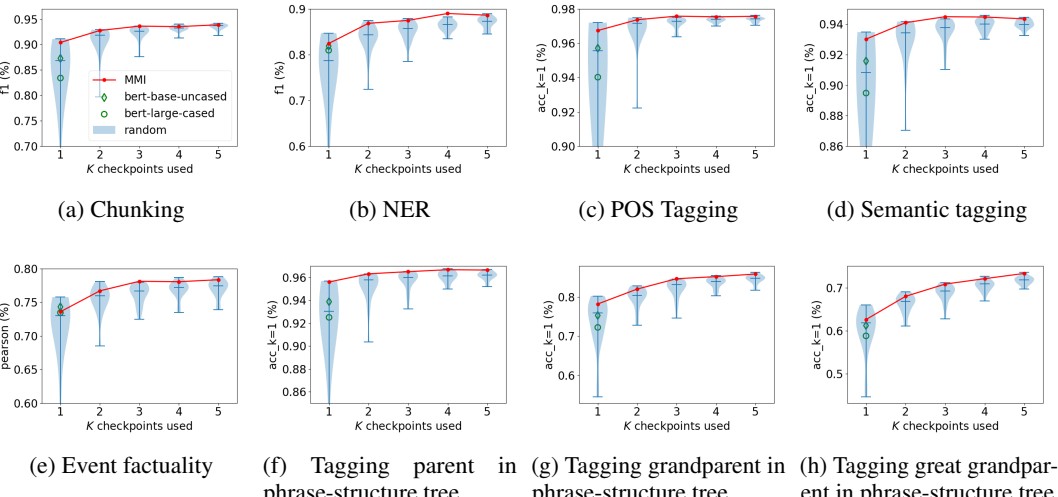

| (a) Chunking | (b) NER | (c) POS Tagging | (d) Semantic tagging |
|---|---|---|---|

| (e) Event factuality | (f) Tagging parent in phrase-structure tree | (g) Tagging grandparent in phrase-structure tree | (h) Tagging great grandparent in phrase-structure tree |
|---|---|---|---|

Figure 3: Performance on a suite of probing tasks: MMI achieves near optimal performance, compared with random search (34 trials at $K = 1$ and 20 for $K \geq 2$). In most of these tasks, the top layer of bert-base-uncased and bert-base-cased are inferior to the first pick by MMI. Surprisingly, in many cases, the top layer of bert-base-cased is worse than half of the 34 checkpoints.

## 5.1 Linguistic Tasks Using Huggingface Checkpoints

We run algorithm 1 on the $\kappa$ matrix estimated in section 3.3. As $K$ increases from 1 to 5, the $\mathcal{S}$ set incrementally includes roberta-base, distilbert-base-uncased, t5-base, bert-base-cased and bart-large. These checkpoints solve multiple tasks: masked language modeling, text-to-text generation, next sentence prediction, and text denoising. Referring to the visualization of $\kappa$ matrix (left pannel of fig. 1), we find that the selected checkpoints all belong to the big cluster in the lower right block. This matches our intuition, as the checkpoints outside this cluster are very isolated.

Then, we apply the selected checkpoints to a suite of probing tasks collected in [28]. To be self-contained, we briefly describe the tasks and datasets.
(a) **Chunking** examines spans and boundaries for each word. We use CoNLL2000 shared task dataset [31].
(b) **NER** predicts the entity type for each word. We use CoNLL2003 shared task dataset [32].
(c) **POS Tagging (POS)** checks if word representations can capture basic syntax. We use PTB dataset [33].
(d) **Semantic Tagging (ST)** is interested in identifying the semantic role in context. The original source of dataset can be found in [34].
(e) **Event Factuality(EF)** aims at determining if an event is factual. We use ItHappened-v2 dataset [35]. Events are given scores between -3 (non-factual) and 3 (factual). For instance, in "I decided not to go.", the word "decided" has a score of 3 whereas "go" has -3. We use Pearson correlation to compare predicted scores against groundtruths.
(f-h) **Syntactic constituency ancestor tagging**: A set of tasks that aim at predicting the constituent label for (f) **parent**; (g) **Grand parent (Gparent)**; and (h) **Great Grand parent (GGparent)** of a word in the phrase structure tree. The structure is derived from PTB.

Following [28], we train a softmax on top of the combined word representations ($\otimes_{i \in \mathcal{S}} \mathbf{f}_i$) for each task. The gradients are not back-propagated through the checkpoints. There are two reasons why we choose to do so. First, for an easier comparison with [28]. Second, there are cases where checkpoints are published as blackboxes, and back-propagation is not allowed. One prominent example is GPT-3 [36]. Although we do not have access to GPT-3 in this paper, we think that our experiment design should respect this fact. Another design choice is that $\mathbf{f}_i$ is taken to be the feature at top layer of the checkpoint. We do not use feature at intermediate layers, again to respect the fact of blackboxes.

The baseline to compare against is random selection. In specific, let $\mathcal{R} \subseteq \mathcal{Z}$ be a set of randomly picked checkpoints, where $|\mathcal{R}| = K$. The task specific head is trained on top of $\otimes_{i \in \mathcal{R}} \mathbf{f}_i$. For $K = 1$,

Table 2: Compare MMI's selection (Algorithm 1, $K = 5$) against a strong baseline [28], which uses the best layer of bert-large-cased for each task. Note that the best layer changes from task to task. It has to be found by brute-force training and evaluation for all 24 layers.

| | chunking | NER | POS | ST | EF | parent | Gparent | GGparent |
|---|---|---|---|---|---|---|---|---|
| MMI selected ($K = 5$) | **93.88** | **88.64** | **97.57** | **94.35** | **78.35** | **96.64** | **85.85** | **73.32** |
| bert-large-cased best layer [28] | 93.64 | 84.44 | 96.73 | 93.83 | 76.25 | 96.10 | 82.46 | 67.90 |

$\mathcal{R}$ ranges over each single checkpoint in the total 34. For $K \geq 2$, we run 20 random picks using different random seeds. Then, the metrics from these random runs can be visualized by violin plots (fig. 3). In each violin plot, the bars on the top and middle indicate best and median performances respectively. We observe an increasing trend in the metrics as $K$ increases from 1 to 5. Importantly, MMI achieves near optimal performance compared with random search, especially when $K > 1$. Note also that random selection results in large variance in performance. Another interesting baseline is a single checkpoint that are commonly adopted. We plot bert-base-uncased and bert-large-cased (at $K = 1$). Surprisingly, for most tasks, bert-large-cased is inferior to half of the 34 checkpoints. This echos some discoveries made in [28]. That is, higher layers of bert model is less transferable to new tasks. Instead, stronger transferability is found in middle-layer features.

Following the above discussion, we collect metrics reported in [28] for the best layer of bert-large-cased. Since layers capture different levels of information [37], the best layer is often task dependent. It has to be found by brute-force training and evaluation for each task. Tab. 2 lists these metrics. Also reported are metrics achieved by $\otimes_{i \in \mathcal{S}} \mathbf{f}_i$ at $K = 5$. We observe that MMI outperforms the best layer of bert-large-cased for most of the tasks, and often wins by a notable margin. This is remarkable, as the checkpoints are selected before seeing any new task, and fixed throughout all tasks.

## 5.2 Image Classification Tasks on Cifar100

We are motivated by applications in computational linguistics. However, we find that the proposed framework extends to computer vision tasks as well. In this section, we simulate an example using cifar100 dataset.

We create $n = 100$ "seen" tasks. Each task is a 30-way classification problem. The 30 classes are randomly sampled from the total 100 classes. Each task has 480 (=500-20) training samples per class. There are 20 training samples held out for each class. The reason will be clear soon. A resnet-50 is trained for each of these "seen" tasks, stored as a checkpoint. Then, we create 20 "unseen" tasks in the same way, but each of them has only 10 training samples (part of the aforementioned holdouts) per class. This small training data is to manifest the transferability of the selected checkpoints. Finally, the remaining 10 holdouts per-class are used as probing data to estimate $\kappa$. We augment this small set of probing images by applying rotations of $\pm 5°$, $\pm 10°$ and $\pm 15°$ degrees.

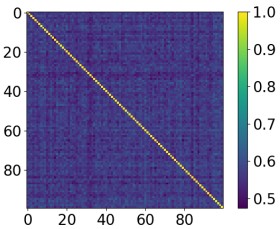

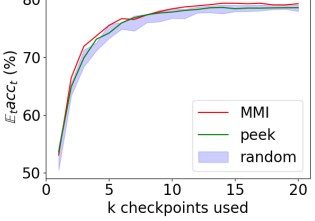

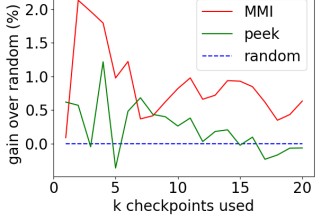

Figure 4: Covariance $\kappa$ for the 100 seen tasks

Figure 5: accuracy averaged over the 20 unseen tasks

Figure 6: Average (over 20 new tasks) improvements over random picks

Figure 4 shows the estimated $\kappa$ for the 100 seen tasks. Compared with the $\kappa$ in fig. 1, it is less structured. That is, tasks tend to be equally distant from each other. This probably suggests that the

space of vision tasks fundamentally differs from that of linguistic tasks. We then apply algorithm 1 on the $\kappa$ to select up to $K = 20$ checkpoints. For each task $t$, a classification head is learned using its $10 \times 30$ training data. The performance of this task is measured by accuracy on the standard validation data (excluding classes not handled in this task), denoted as $acc_t$. As before, a baseline is random selection. In addition, suppose we could peek at the first unseen task. We may identify the top-$K$ best checkpoints for this task, by running training and evaluation. Then we can keep using these top-$K$ checkpoints for all new tasks. This gives us another method to compare against, dubbed "peek".

In fig. 5, we plot, $\mathbb{E}_t acc_t$, the average validation accuracy over tasks against $K$. The shaded area are accuracies of runs using different random seed. Overall, as expected, the accuracies increase and saturate at larger $K$. To have a better view, we plot the gains of MMI and "peek" against random. That is, the MMI (and "peek") curve subtracting the average curve of the shaded area in fig. 5. The gains are shown in fig. 6. MMI steadily outperforms the random baseline. Whereas "peek" fluctuates and can be inferior than random at multiple $K$'s. This may be caused by "overfitting" to the first task that it peeked, which prevents generalization to other tasks. In analogous to senor placement, this amounts to optimizing the placement for one location, which could harm the accuracies for other locations. In contrast, MMI does not exhibit this issue.

## 6 Conclusion

In this paper, we motivate a problem that addresses two recent facts: 1) huge amount of published checkpoints; 2) emerging new tasks. By drawing analogy from optimal sensor placement, we propose a checkpoint selection framework without seeing any new task. The key steps are: 1) Estimating task relevance with proprietary constraints (e.g., data, task-specifc heads); 2) selection by maximizing mutual information, assuming a Gaussian Process on the tasks. Effectiveness is validated on tasks from computational linguistics as well as computer vision. However, it should be reminded that there are a few assumptions (section 3.2) made for step 1). We partially validated them on text data (section 3.3 and 3.4). A more comprehensive validation on other data (e.g., images) is left for future work. Aside from that, it is also interesting to explore more fine-grained models of the task space (e.g., hierarchical) and experiment with more tasks.

## Acknowledgement

The authors thank Nelson Liu for helping on data of linguistic experiments. The first author thanks Yifan Wei for her company and support during the pandemic lockdown.

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
