# Supplementary Material

## Proof of Proposition 1

Denote $\text{vec}(\cdot)$ vectorization of a matrix. It follows that

$$\kappa(i,j) \triangleq KA(\mathbf{F}_i, \mathbf{F}_j) = \frac{\langle \text{vec}(\mathbf{F}_i^\top \mathbf{F}_i), \text{vec}(\mathbf{F}_j^\top \mathbf{F}_j) \rangle}{\|\text{vec}(\mathbf{F}_i^\top \mathbf{F}_i)\| \cdot \|\text{vec}(\mathbf{F}_j^\top \mathbf{F}_j)\|},$$

which is inner product between normalized $\text{vec}(\mathbf{F}_i^\top \mathbf{F}_i)$ and $\text{vec}(\mathbf{F}_j^\top \mathbf{F}_j)$. Hence $\kappa$ is the Gram matrix of

$$\begin{bmatrix} \frac{\text{vec}(\mathbf{F}_1^\top \mathbf{F}_1)}{\|\text{vec}(\mathbf{F}_1^\top \mathbf{F}_1)\|} & \cdots & \frac{\text{vec}(\mathbf{F}_n^\top \mathbf{F}_n)}{\|\text{vec}(\mathbf{F}_n^\top \mathbf{F}_n)\|} \end{bmatrix},$$

which is PSD.

## Proof of Proposition 2

Recall the definition: a set function $f(\mathcal{S})$ is submodular, if for any subsets $\mathcal{S} \subseteq \mathcal{S}' \subseteq \mathcal{Z}$, and $i \in \mathcal{Z} - \mathcal{S}'$,

$$f(\mathcal{S} \cup \{i\}) - f(\mathcal{S}) \geq f(\mathcal{S}' \cup \{i\}) - f(\mathcal{S}').$$

Referring to Eq. (3), we realize that the left side equals $H(\{i\}|\mathcal{S}) - H(\{i\}|\bar{\mathcal{S}}_i)$, and the right side equals $H(\{i\}|\mathcal{S}') - H(\{i\}|\bar{\mathcal{S}}_i')$. Since conditioning on more variables reduces entropy, we have $H(\{i\}|\mathcal{S}) \geq H(\{i\}|\mathcal{S}')$ and $H(\{i\}|\bar{\mathcal{S}}_i) \leq H(\{i\}|\bar{\mathcal{S}}_i')$. It therefore holds that $H(\{i\}|\mathcal{S}) - H(\{i\}|\bar{\mathcal{S}}_i) \geq H(\{i\}|\mathcal{S}') - H(\{i\}|\bar{\mathcal{S}}_i')$.

## Tasks that Huggingface Checkpoints were Trained on

1. **albert-base**, **albert-large**: masked language modeling + sentence order prediction

2. **bart-base**, **bart-large**, **bart-large-cnn**: text denoising

3. **bert-base-cased**, **bert-base-uncased**, **bert-large-cased**, **bert-large-uncased**: masked language modeling + next sentence prediction

4. **distilbert-base-cased**, **distilbert-base-uncased**, **distilbert-base-multilingual**: knowlededge distillation on bert (matching representatations of bert)

5. **gpt**, **gpt2**, **gpt-medium**, **gpt-large**: causal language modeling

6. **longformer-base**: masked language modeling

7. **roberta-base**, **roberta-large**: masked language modeling

8. **roberta-large-mnli**: masked language modeling + entailment (finetuned on MNLI dataset)

9. **t5-3b**, **t5-base**, **t5-small**, **t5-large**: text-to-text generation

10. **xlm-clm-ende-1024**, **xlm-mlm-100-1280**, **xlm-mlm-17-1280**, **xlm-mlm-ende-1024**, **xlm-mlm-enfr-1024**, **xlm-mlm-enro-1024 xlm-roberta-base**, **xlm-roberta-large**: crosslingual masked language modeling

11. **xlnet-base-cased**, **xlnet-base-large**: permutation language modeling

35th Conference on Neural Information Processing Systems (NeurIPS 2021), Sydney, Australia.

## Details on Training

**For experiments in section 5.1**, we use a batch size of 32 sentences, adam optimizer with a learning rate of 1e-3. We run for 40 epochs and report the test metric at the "best" validation epoch.

**For experiments in section 5.2**, all checkpoints are instances of resnet-50. They are trained by a batch size of 128, and an initial learning rate of 0.1. We run for 200 epochs, with learning rate decay at the 60th, 120th and 160th epoch. A typical validation accuracy from these checkpoint (on its own task) is about $83\%$ (reasonably good). For the 20 new tasks, we experiment with a softmax classifier on top of selected checkpoints. The learning rate is kept at 0.1. We report the best validation accuracy for each of the 20 tasks. For each task, its validation set is standard cifar100 validation split, but only includes the classes that are involved in this task.


# References

[1] N. F. Liu, M. Gardner, Y. Belinkov, M. E. Peters, N. A. Smith. Linguistic Knowledge and Transferability of Contextual Representations. Proceedings of the 2019 Conference of the North American Chapter of the Association for Computational Linguistics: Human Language Technologies. 2019.