# OpenReview forum: "Exploiting a Zoo of Checkpoints for Unseen Tasks"
_NeurIPS.cc/2021/Conference — NeurIPS 2021 Poster_

### Official Review · Reviewer_YBj2 · 2021-07-07

**Rating:** 6
**Confidence:** 3

**Summary:**


The paper considers the question of how to select an ensemble of existing pretrained models to be used as feature extractors, when the task for which they are to be used is not known a priori. It seems to aim less to generalize to a given fixed target task distribution, but rather at generalizing over an expectation over target task distributions.

To achieve this, the authors propose essentially two steps:

Firstly, considering the feature representation of each model from the zoo, and computing a within-model similarity matrix between the features extracted for one fixed model over a set of samples. Then to use the cosine angle between similarity matrices of different models as a similarity between models.

Secondly, to select a fixed number of models (corresponding to a fixed budget) based on mutual information estimates between the set of models and the remaining models. This is intuitively appealing.

**Ethical Concerns:**

The reviewer sees no issues regarding ethics.

**Ethics Review Area:**

["I don’t know"]

**Limitations And Societal Impact:**

As stated in the previous box:

While theoretically interesting, what is the practical applicability of optimizing for an expectation over tasks and not for a given single target task ? When this can be helpful (as opposed to optimizing for a given target task which one usually has in practice)?
The reviewer sees this as some level of weakness and feels that motivating this would make the paper stronger.

The discussions on submodularity yielded no conclusive result and are maybe not the most important.

The reviewer sees no issues regarding societal impact. Reusing checkpoints is positive, as it does  waste less GPU training time and energy.

**Main Review:**

Originality: the approach seems to be novel. The reviewer finds the amount of originality good enough for NeurIPS.

Quality:

there are limitations in the experiments:

1 lack of comparison against existing baselines for ensemble selection, in particular A-Map seems to be within the scope of the question.

They compare only against random selection for NLP and random and top-k performance selection for CIFAR-100. At least for CIFAR computing gradients for A-MAP as another baseline should be not an issue.

possibly they could also use

2a evaluation using a small number of single point tasks in 5.1. While they seem to optimize without having a fixed task in mind, they evaluate on four standard tasks, for which one may suspect that some of the NLP models were implicitly optimized for and thus possibly leaking onto the 4 tasks.

2b As a consequence, when evaluating against 4 fixed tasks as in 5.1, one should also show how the best single model selected for one task using a task-dependent selection algorithm as baseline performs ... on all 4 tasks (including also the three tasks for which it was not optimized).

The reviewer disagrees on the statement that backprop is expensive when discussing A-MAP. Each model was trained using backprop, thus it seems to be feasible to use backprop also after training, at least until the feature representation if the computations below were a black box.


Significance: In terms of setup there is one doubt: While theoretically interesting, what is the practical applicability of optimizing for an expectation over tasks and not for a given single target task ? When this can be helpful?

Question: 5.2 - how often this experiment has been repeated?


Clarity, the paper is written such that it can be read easily.
However, a few things can be improved:

3.2 the interpretation of the kernel alignment in terms of cosine-angle between similarities matrices within a model can be added. This interpretation also holds without any assumptions on the tasks

Sec 2: generalizing to unseen tasks seems to amount to optimizing over an expectation of a statistic over tasks. This can be worked out somewhat.

Is there a connection between optimizing MI over feature similarities between the chosen set and the complement set of tasks, and under certain assumptions, between the expected prediction performance over many tasks ? It might be helpful to assume that each model and its feature representation is optimal for the task it was trained for, among the set of all models used for MI optimization.
The reviewer would be surprised if such a link could not be discussed (whereas the submodularity does not give a clear result, this would provide some theoretical link to the set of a priori unknown target tasks).


typos:

This objective is submoludar.

how many tasks has to be seen
--> have

 picking checkpoints amount to identifying useful
--> amounts

In optimal senor placement [15]



**Time Spent Reviewing:**

3.5 hours

---

> ### Author Response · Authors · 2021-08-10
> **Addressing all questions**
>
> We thank the reviewer for evaluating our paper as novel and original. We address all questions below.
>
> 1. "comparison against A-map"—It is a very interesting experiment. However, extending A-map to language models (the main motivating application of this paper) is not straightforward. In particular, let two checkpoints under comparison be $\mathbf f_1(\mathbf x)$ and $\mathbf f_2(\mathbf x)$. A-map considers $cos(\nabla \mathbf f_1(\mathbf x), \nabla \mathbf f_2(\mathbf x))$ as their similarity. Here $\mathbf x$ is (and must be) a shared input. For computer vision applications, it is straightforward to just input the same probing image as $\mathbf x$. However, for language models, $\mathbf x$ is word embedding. For two language models under comparison, even the same text input can correspond to different word embeddings, as the two language models usually have different look-up embedding layers. Fully addressing this difficulty is beyond the scope of this paper, and may deserve another separate paper. We will highlight these facts in final version.
>
> 2. Addressing 2a-2b. First, a small correction: there are 6 NLP tasks, instead of 4. The suggested task-dependent selection is an interesting case to consider. We ran experiments to show that simply selecting the best checkpoint for one task would not necessarily guarantee the best performance for all tasks. For example, “albert-base” turns out to be the best checkpoint for task “Parent”. However, on most of the other tasks, it is inferior to Roberta-base (the 1st pick by max mutual information). Details are in the table below.
> ```
> checkpoint   |  POS    |   ST    |  GED   |  Parent | GParent | GGParent
> albert-base  |  96.31  |  92.83  |  34.81 |  95.66  | 78.61   | 62.14
> roberta-base |  96.71  |  93.06  |  35.28 |  95.60  | 78.19   | 62.60
> ```
>
> 3. "statement that backprop is expensive when discussing A-MAP" --- We agree. one back-prop is not that expensive. We will revise that to “moderately more expensive than a single forward pass”.
>
> 4. "practical applicability of optimizing for an expectation over tasks", also addressing the concern in limitations and social impact --- With so many published checkpoints, people may wonder what checkpoint(s) should be adopted for new tasks. The best checkpoint(s) may vary from task to task. A straightforward approach is to run brute-force training and evaluate using each checkpoint (and combinations), for each target task. This is very costly. An alternative approach is to just adopt popular checkpoints, like BERTs. However, as shown by fig. 3, they do not necessarily achieve the best performance. To address this challenge, the proposed method provides a strong baseline with modest computational cost. We believe this will benefit a practitioner who wants a quick estimate of ballpark performance for a particular task. We will add the above remarks to the final version.
>
> 5. "How often is the experiment in section 5.2 repeated?" --- We repeated for 20 times with different random seeds, and fig. 4 is obtained by averaging gains from each run.
>
> 6. On clarity issues:
>
>    For section 3.2, we will add interpretation of kernel alignment as suggested.
>
>    For section 2, we will highlight that
>    $$\min_{\mathcal S} \mathbb E_{p_t\sim \mathcal P}[\mathbb E_{(\mathbf x, y)\sim p_t}[\ell_t(\otimes_{i\in\mathcal S} \mathbf f_i (\mathbf x), y)]]$$
> is the objective, where $\ell_t$ is the loss for task $t$.
>
>    For a theoretical discussion of the relationship between mutual information (among tasks) and the above objective: Invoking infomax principal, we in fact want to $$\max_{\mathcal S} \mathbb E_{p_t\sim \mathcal P} I(\otimes_{i\in\mathcal S} \mathbf f_i(\mathbf x); y), \mbox{ where } (\mathbf x, y)\sim p_t(\mathbf x, y),$$
> and $\mathcal S$ indexes "seen" tasks. A finite sample approximation of the above is
> $$\max_{\mathcal S} \frac{1}{n}\sum_{j=1}^n I(\otimes_{i\in\mathcal S} \mathbf f_i(\mathbf x); y_j),$$
> where $y_j$ denotes the label for $j$-th task. As suggested by the reviewer, it is then helpful to assume $\mathbf f_j(\cdot)$ to be a “perfect” feature extractor for any $j$-th task. In other words, for any $j$-th task, $\mathbf f_j(\mathbf x)$ contains all information of $y_j$. We then argue to use
> $$\max_{\mathcal S} I(\otimes_{i\in\mathcal S} \mathbf f_i(\mathbf x); \otimes_{j=1}^n \mathbf f_j(\mathbf x))$$
> as a surrogate. If we further exclude seen samples from $\otimes_{j=1}^n \mathbf f_j(\mathbf x)$, that leads to
> $$\max_{\mathcal S} I(\otimes_{i\in\mathcal S} \mathbf f_i(\mathbf x); \otimes_{j\in\bar{\mathcal S}} \mathbf f_j(\mathbf x)),$$
> which is Eq (2) in the main paper. We will add this discussion to the final version. Nevertheless, a full theoretical development is beyond the scope of this paper right now, and we defer that to future work.
>
>    On the other hand, the discussion on submodularity is purely from an optimization perspective, i.e., closeness between the proposed greedy method and a brute-force search that exactly maximizes mutual information. As suggested, we will reduce the content in this section.
>
> 7. Typos will be corrected.

---

> > ### Comment · Reviewer_YBj2 · 2021-09-01
> > **Thank you for the rebuttal**
> >
> > The authors addressed all points.
> > 1. As for A-map that would be still an interesting result to be seen.
> > 2. it is a good outcome
> > 4. That is a reasonable reply, means one can say it is useful when one is on a computational budget and wants a choice by prior guess.
> > all the others are satisfactory.
> > The reviewer feels it is a paper above acceptance threshold though not on the level of a strong result like score 8 or 9 .

---

### Official Review · Reviewer_8waw · 2021-07-12

**Rating:** 6
**Confidence:** 4

**Summary:**

provide a method of how to (adaptably) combine  existing task models for new unseen task with aiming to relieve practitioners'  selection difficulty.  Through analogizing sensor placement problem, the authors give the above method and conduct effective verification on language and CV tasks.

**Limitations And Societal Impact:**

Considering both data biases and task biases, the proposed method inherits naturally these biases to some extent.

**Main Review:**

The idea of the submission is interesting and it uses the sensor placement as checkpoints to formulate the problem in hand for intuitive understanding. Concretely, using GP to model task space to capture the task relation and while using checkpoints and unlabeled probing data to estimate covariance of the GP involved, then using Max mutual information criterion to identify the representative checkpoints for new tasks. To my best knowledge, this is relatively new. the experiments also show effectiveness of the proposed method. My comments are
1. a task corresponds to its data distribition, then what differences are between the distribution/task-oriented inductive learning and the proposed method?
2. when a task is viewed a domain, again what differences are between the domain generalization and the proposed method?
3. the checkpoint selection here is quite similar to the coreset selection, thus in some sense this point is not new.
4. if the selected checkpoint diversity is needed, or can the diversity be useful?

**Time Spent Reviewing:**

>4hr

---

> ### Author Response · Authors · 2021-08-10
> **Addressing all questions**
>
> We thank the reviewer for evaluating our paper as interesting and new. Below, we answer all questions in the order they are raised.
>
> 1. Distribution/task-oriented inductive learning requires learning from scratch, for both the feature extractor and task-specific “head”. The proposed method allows one to warm-start from existing strong feature extractors, and only a task-specific “head” is learned from scratch.  Therefore, it is much less data-hungry.
>
> 2. Domain generalization often knows what target task to optimize for, and then tries to bridge the gap between the source and target domain. However, this paper is interested in the case when the target tasks are not known a priori.
>
> 3. Coreset selection finds a subset that well represents the full set. Recently, it is often applied to thinning of data samples for efficient training. Instead, this work addresses selection of (seen) tasks by maximizing a sub modular function. The goal is better generalization.  Hence, we are attacking a different problem. We will highlight these facts in the introduction in final version.
>
> 4. In fact, the diversity has been accounted by the max mutual information objective. That is, the first term of the last line in Eq. (3) requires  one to pick the next checkpoint such that it diversifies the current chosen set. More discussion can be found on lines 236-238. The proposed strategy encourages diversity, and experiments validate its usefulness.

---

> ### Comment · Reviewer_8waw · 2021-08-26
> **Thanks for response**
>
> The authors have made the responses to all the reviewers. I also read these. One the whole, The idea is indeed interesting for me but does not give me some reason of increasing the score sufficiently.

---

### Official Review · Reviewer_zpV4 · 2021-07-18

**Rating:** 6
**Confidence:** 4

**Summary:**

When no task specific heads are available sample labels cannot be predicted and the dependency between two tasks and thus task transferability cannot me measured. The paper uses a Gaussian process to model the task space and comes up with a closed solution for the mutual information gain. Proposes a greedy selection algorithm for selecting checkpoints to maximize the information gain. Experiments with text and image datasets are performed to show that selecting checkpoint based on this criterion outperforms random selection.

**Ethical Concerns:**

None.

**Limitations And Societal Impact:**

Yes.

**Main Review:**

Strengths:
-	Using unsupervised probing to get around the problem of missing true and predicted labels and showing that kernel alignment can be achieved reasonably well at the level of feature extractors without having access to task specific heads.
-	Good leveraging of the analogy between sensor replacement and checkpoint selection for a more generalizable task transfer from seen tasks to unseen ones.
Weaknesses:
-	Imaging experiments are less compelling


Each checkpoint is added in a greedy way to maximize the gain, where the gain is formulated as the information gained by adding the current checkpoint to the current set of checkpoints minus the information lost by removing the current checkpoint from the remaining part of the checkpoint zoo. A good checkpoint generalizable to unseen tasks is the one that is representative of the remaining part of the checkpoint zoo (the information loss minimum when removed) but at the same that surprises the current set of checkpoints the most.

In the CIFAR 100 experiments classes within seen and unseen tasks can overlap as 30 classes per task are chosen randomly. What would happen if unseen tasks do not contain any classes used in seen tasks?

Imaging experiments select n=100 seen tasks. How does the accuracy change with respect to n?

It is not clear why choosing too many checkpoints hurts the performance in such a significant way? Overfitting explanation is not very compelling

Typos:
From the abstract: “This objective is submoludar”. Is this supposed to be submodular?
An extra “the” in line 225
Is missing in line 232.
Senor placement lince 340


**Time Spent Reviewing:**

5 hours

---

> ### Author Response · Authors · 2021-08-10
> **Addressing all questions**
>
> We thank the reviewer for the constructive comments, and answer all questions below.
>
> 1. “What would happen if unseen tasks do not contain any classes used in seen task” – We have experimented with this case before submission. Results are not included in the submission due to page limit. In this case, all accuracies are lower. However, the trend in fig.4 still holds.  That is, the setup is same as in the paper, except that the unseen tasks cover 50 classes, and seen tasks cover the other 50.  As the number of picks, $K$, ranges from 1 to 20, random selection’s accuracy increases from 40.08% to 52.46%.  In contrast, the max-mutual-info selection’s accuracy increases from 44.30% to 52.90%.  There are stable gains against the baseline, and we will include more discussions in final draft.
>
>
> 2. “How does the accuracy change with respect to $n$, the number of seen classes”— When $n$ is larger, we have a bigger pool of seen tasks, and a bigger set of checkpoints to choose from.  The overall accuracy increases.  For example, suppose we keep the same setup as in the paper, but increase $n$ from 100 to 200.  The following are representative accuracies at different $K$ (number of picks). Overall, accuracies for $n=200$ are higher than those for $n=100$. But again, MMI selection shows stable gain over random selection.
> ```
>                           | random (K=1)    | random (K=20) | MMI (K=1)  | MMI (K=20)
> n=100 (in paper)       |  52.98%         | 78.63%        | 53.07%     | 79.26%
> n=200                  |  54.20%         | 78.87%        | 55.85%     | 79.13%
> ```
>
>
> 3. “why choosing too many checkpoints hurts the performance. Overfitting explanation is not very compelling”—We want to clarify this. For any method in fig.4, its accuracy always increases as more checkpoints (bigger $K$) are included. See the table for question 2. However, the gain against random selection diminishes at big $K$ (fig. 4), as all accuracies saturate in that case. This is especially true fo  the top-K-“peeking” method.  At bigger $K$, the “peeking” method selects too many checkpoints that are just good for the task it peeked, lacking generalizability to other unseen tasks. We agree the word “overfitting” is confusing, and will clarify that in final version.
>
> 4. Yes, it should be "submodular". We will correct all typos as suggested.

---

> > ### Comment · Reviewer_zpV4 · 2021-08-28
> > **Thanks for clarification**
> >
> > Thank you for the additional clarification and for running additional experiments in a short time. Although the responses help alleviate some of my concerns, regrettably not at a level to improve my original rating.

---

### Decision · Program_Chairs · 2021-09-27

**Decision:**

Accept (Poster)

**Comment:**

PRELIMINARY: The reviewers generally agree that the paper is interesting and can be accepted for publication at NeurIPS.